# Feature-Based Analysis of Theory of Mind Representations in Neural Network Models

## Abstract

Theory of Mind (ToM) presents a significant generalization challenge in computational modeling. This paper explores how neural networks with varying architectures and training regimes learn and represent ToM-related features. We introduce a novel method for quantifying feature representation within neural networks and apply it to a set of theoretically-grounded features designed to differentiate between hypothesized ToM strategies. We examine the relationship between feature representation and task accuracy across different model architectures and training datasets. This work provides insights into the mechanisms underlying ToM capabilities in neural networks and offers a framework for future research in computational ToM.

The study of Theory of Mind (ToM)—the ability to attribute mental states to others—began with Premack and Woodruff's work on chimpanzees in 1978 Premack & Woodruff (1978). Since then, it has grown into a fundamental concept in cognitive science. Over the past five decades, developmental and comparative psychologists have proposed various theories to explain how humans and non-human animals acquire and use ToM skills. These theories have sparked heated debates and led to numerous experimental paradigms, from false-belief tasks to nuanced measures of mental state attribution. However, traditional behavioral studies are limited by methodological constraints and the inherent difficulty of inferring cognitive processes from observable behavior alone.

Recent advancements in computational modeling provide new tools for simulating and analyzing ToM processes in controlled, replicable environments. These simulated models offer unique insights into the cognitive mechanisms underlying ToM, complementing traditional behavioral studies. However, researchers continue to face challenges in creating models that generalize learned skills to novel scenarios, mirroring difficulties observed in non-human primate studies.

Psychologists have proposed a spectrum of theories to explain ToM task performance in human and non-human subjects, ranging from simple behavior reading (no mentalizing) to more powerful belief-desire mentalizing. There are ongoing debates about whether different strategies are capable of solving certain ToM tasks as well as robustly generalizing to new tasks. Computational models like neural networks can be used to experimentally study how an agent might learn from experience to do ToM tasks. Without analyzing the features represented within these models, we cannot reliably differentiate between these strategies in neural networks, impeding our ability to implement and study mechanisms and curricula that foster genuine ToM capabilities and robust generalization.

In this paper, we:

- Develop and validate a novel technique for quantifying the implicit representation of features within neural networks' activations, providing a robust method for analyzing internal model representations.

- Describe a comprehensive set of theoretically-grounded features to empirically evaluate our feature extraction method and differentiate between three hypothesized ToM strategies in neural networks

- Quantify the feature representation capabilities of diverse neural network architectures and training regimes, offering insights into their respective ToM capabilities and potential underlying mechanisms

- Investigate the relationship between feature representation and model performance, assessing the necessity and sufficiency of specific features for accurate task completion

# 1 RELATED WORK

## 1.1 COMPUTATIONAL THEORY OF MIND

This work is heavily inspired by the ToMnet experiments of Rabinowitz et al. Rabinowitz et al. (2018). In their study, they implement machine learning models with explicit ToM-like representations about agents' attributes and mental states, and are able to leverage the computational setting to probe those models for representations of those features.

Recently, Horschler et al. Horschler et al. (2023) used computational modeling to investigate ToM capabilities in non-human primates, focusing on visual perspective-taking tasks similar to the one investigated by this paper. They developed seven models of varying complexity to represent different theories of primates' social cognition, and parameterize the subjects' reliance on their ToM inferences to determine how well the theories explain primate behavior.

Computational ToM skills have also been particularly well-studied recently in the context of large language models (LLMs). The ToMi dataset by Le et al. Le et al. (2019) consists of short, structured narratives based on the Sally-Anne false belief test. ToMi focuses primarily on first-order ToM reasoning about physical world states. More recently, Xu et al. developed OpenToM Xu et al. (2024) to benchmark ToM capabilities in large language models using longer, more natural narratives, covering both physical and psychological aspects of ToM. Despite recent advancements, LLMs continue to significantly underperform humans on ToM tasks. The significant gap between LLMs and humans on such tasks highlights the difficulty in acquiring robust ToM skills in machine learning models.

## 1.2 HYPOTHESES ABOUT TOM IN NONHUMAN ANIMALS

Developmental and comparative psychologists have produced a variety of theories about potential ToM mechanisms. In this section we briefly overview some theories about ToM in developing humans and non-human animals. This list is incomplete, and much work remains to be done regarding the extent to which any of these theories can be (and/or have already been) used to generate specific cognitive hypotheses, which might be implemented computationally and tested empirically. We also emphasize the importance of noting the context in which these theories were generated, and which scientific problems they address.

Simulation theory involves using one's own mental states as a model for replicating or simulating those of others. It could be facilitated by mirror neurons, which are thought to enable the internal mirroring of others' actions and emotional states. Additionally, it emphasizes the role of pretend play during childhood development (Gallese & Goldman, 1998), (Harris, 1992). Theory theory involves updating (often Bayesian) beliefs based on empirical evidence about others. Children refine their understanding by testing internal models empirically during social interactions (?). Mentalizing/systemizing theory differentiates between systematizing, reasoning about non-social rules, and mentalizing, reasoning about social rules (Baron-Cohen, 2000). This theory has been used to explain differences in the ToM skills of individuals on the Autism spectrum, who might tend to systemize more strongly relative to mentalizing.

Of particular relevance to the work in this paper are theories about ToM in animal cognition, as reviewed and categorized by Heyes Heyes (2015). Heyes outlines several key conceptual frameworks for understanding potential mindreading abilities in animals. Under behavior reading, animals predict others' behavior based solely on observable cues, without attributing mental states Heyes (1998). Perception-goal psychology is a framework proposed by Call and Tomasello Call & Tomasello (2008) as an alternative to full-blown ToM in animals. It suggests that great apes may understand what others perceive and what concrete goals they may have, without attributing abstract mental states like beliefs. Minimal ToM is a more recent framework by Butterfill and Apperly Butterfill & Apperly (2013), who propose a limited form of mindreading that allows for some attribution of mental states, such as goals and perceptions, but without full metarepresentation of propositional attitudes. Whiten's intervening variable approach Whiten (2013) views animal mindreading as analogous to how comparative psychologists use intervening variables to explain animal behavior. Finally, full-blown ToM involves the ability to attribute complex mental states, including false beliefs Premack & Woodruff (1978).

## 2 TEST ENVIRONMENT

In this paper, we use the Standoff test environment, a gridworld setting that replicates the competitive feeding paradigm computationally Michelson et al. (2024). The competitive feeding paradigm is a test setup designed to distinguish whether a non-verbal subject will change its behavior to account for what it believes someone else *knows*, based on evidence relating to what the other person *sees* (Hare et al., 2000). Specifically, the environment implements a version of competitive feeding in the style of Penn and Povinelli Penn & Povinelli (2007). In Standoff tasks, two treats of different sizes are visibly hidden in any of five boxes, which are then shuffled around. The player's challenge is to select the box containing the best possible treat; this is made difficult by the presence of an opponent. The opponent follows simple behavioral rules: if it believes the larger treat is somewhere, it will claim that box, preventing the player from getting whatever treat is inside. Otherwise, the opponent will attempt to reach the smaller treat, or will select a preferred box. Those rules are obfuscated by the fact that the opponent's vision might be obscured at any point during the setup. The opponent might be unaware that either treat exists, or it might harbor a counterfactual belief about either of the treats' locations. The player's best option is always to either stay clear of the opponent or to take advantage of the opponent's unawareness.

We use this setting as a source of data for supervised learning; each datapoint is collected from a single trial of five timesteps, or a (5, 5, 7, 7)-sized video. The target output to be learned is the *correct* box, meaning the player's best choice of the five boxes, given the opponent's selection. Twenty percent of all datapoints are reserved for the evaluations referenced throughout this paper. Previously, Michelson et al. found that models trained on different subsets of Standoff were able to learn the tasks present in their training data to high accuracies, but struggled to generalize to novel settings Michelson et al. (2024).

In this paper, we use three training settings, patterned off of Penn and Povinelli's description of systematic competitive feeding: *Stage-1* has all tasks without an opponent present. *Stage-2* has all tasks in Stage-1, plus those with a fully-informed opponent. *Stage-3* has all tasks allowed in the environment. Since ToM is a generalization challenge, we split many results in this paper by novelty. Note that all tasks are familiar to models trained on Stage-3.

## 3 EXPERIMENT 1: FEATURE EXTRACTION

In this experiment, we demonstrate the process by which we quantify features' presence within neural networks' inner activations.

### 3.1 MODELS

We train neural networks with three different architectures as our task-models. The selected architectures are a multilayer perceptron (MLP), a convolutional neural network (CNN), and a convolutional LSTM (CLSTM). The MLP has two hidden layers of 32 units and ReLU activations. The CNN model has two convolutional layers, followed by a single 32-unit hidden layer. Finally, the CLSTM model has a convolutional layer that is applied to data at each timestep, whose results are fed to a three-layer LSTM with 32 hidden units in each layer. A 32-unit hidden layer is used to produce output predictions from the last timestep of the LSTM's outputs. For each dataset and architecture, we train three task-models with different random seeds for batches and weight initialization. Inputs to the MLP are flattened, and inputs to the CNN stack all five timesteps into the channel dimension.

### 3.2 METRIC: INTRINSIC FEATURE REPRESENTATION SCORES

The models that we train to complete our task (task-models) are connectionist, so we cannot expect to find explicit, conveniently structured representations of discrete features. While methods like correlation analysis or mutual information measurements provide some insight into feature representations, they do not capture the complex, non-linear relationships that exist between perceptron activations and high-level features, making the use of more sophisticated feature extraction techniques necessary for accurate quantification. To quantify the degree to which features are implicitly represented within neural networks' activations, we train additional machine learning models.

These additional extraction-models take as inputs the internal activations of the task-models', collected when the task models are being evaluated. These internal activations might be sourced from all hidden layers, or any specific hidden layer. Using those activations, each extraction-model is trained to predict one of our defined features. Its accuracy in making that prediction is considered the Intrinsic Feature Representation (IFR) score. We use two architectures for extraction-models: linear regression, and a multilayer perceptron with one hidden layer of 32 units. The extraction-model architectures are intended to be simpler than the original model, to ensure that they do not make inferences the original model could not. We assume that, because these models might extract connectionist features in the same manner that a task-model's internal layers might, their ability to predict a given feature corresponds with the task-model's ability to use that feature. Hence, the extraction-models' accuracy scores quantify task-models' feature representations in terms of their usefulness to those models.

We train our extraction-models for 10,000 batches of 64 datapoints each. They are trained using the Adam optimizer and an exponentially decaying learning rate schedule, starting at 0.01 and decreasing by 1% each epoch. Since the extraction models must produce vectors of varying types which may not be one-hot, their loss function is the mean squared error of their outputs. Accuracy is reported by rounding outputs' values to the nearest integer (all features tested in this paper are binary). Each task-model provides data for three differently-seeded extraction-models.

The features of our set might also be accurately inferred from random chance—especially when a feature is dominated by one value—or from the raw perceptual inputs of the environment. To account for these factors, we imitate the IFR scoring procedure using the task-models' inputs, producing raw input IFR scores to compare activation IFR scores against. Activation IFR scores might be lower than raw input IFR scores for several reasons: for example, the connectionist nature of the activations might be more difficult to parse than the discrete inputs, or the neural network might discard information about the given feature before the target layer. When activation IFR scores are higher than the raw input IFR scores, we take that as evidence that the task-model performs useful computations towards better-representing the given feature.

### 3.3 FEATURES

In this experiment, we investigate the IFR scores of two features: *opponents*, and *b-loc*. *Opponents* is a binary scalar that indicates the presence or absence of an opponent, since the environment has at most one opponent in each trial. We expect that all models should represent opponents well, even on novel tasks, because the feature is easily observable in the perceptual input. *B-loc* represents a more complex inference that our models might make: the opponent's ground-truth belief state regarding the location of both treats. This feature is a 10-length vector representing the two treats' potential presence in each of five locations. We expect that *b-loc* might be well-represented on familiar tasks for all models. We do not expect, however, that any models should represent *b-loc* particularly well on novel tasks. Doing so would be evidence for strong ToM skills, and *b-loc* is particularly useful for predicting the opponent's selection across all tasks in the environment.

### 3.4 RESULTS

Raw input IFR scores are strong for *opponents* across all tasks, indicating that the feature is accessible given a sufficiently powerful extractor model. *b-loc*, on the other hand, becomes more difficult to predict with each sequential dataset in the familiar case. The feature extractor generalizes better to novel cases after being trained to predict *b-loc* on S2, which covers more unique *b-loc* possibilities. Although we tested both linear and nonlinear extractors, we find that the nonlinear extractors' performance is universally identical or better than that of its linear counterpart. While linear models might struggle to find data in long vectors, underestimating a feature's representation, nonlinear extractors could potentially make inferences that go beyond the most surface-level observations about the environment and overestimate. We must take that fact into account when comparing our feature extractors' performance across different inputs.

From these results, we can see a clear difference between the two features tested: all models' representations of *b-loc* are much less robust to novel tasks than their representations of *opponents*. This finding is unsurprising, because novel tasks often introduce unseen *b-loc* values as models transition from Stages 1 and 2 to 2 and 3. Although the CLSTM model does not generalize particularly well,

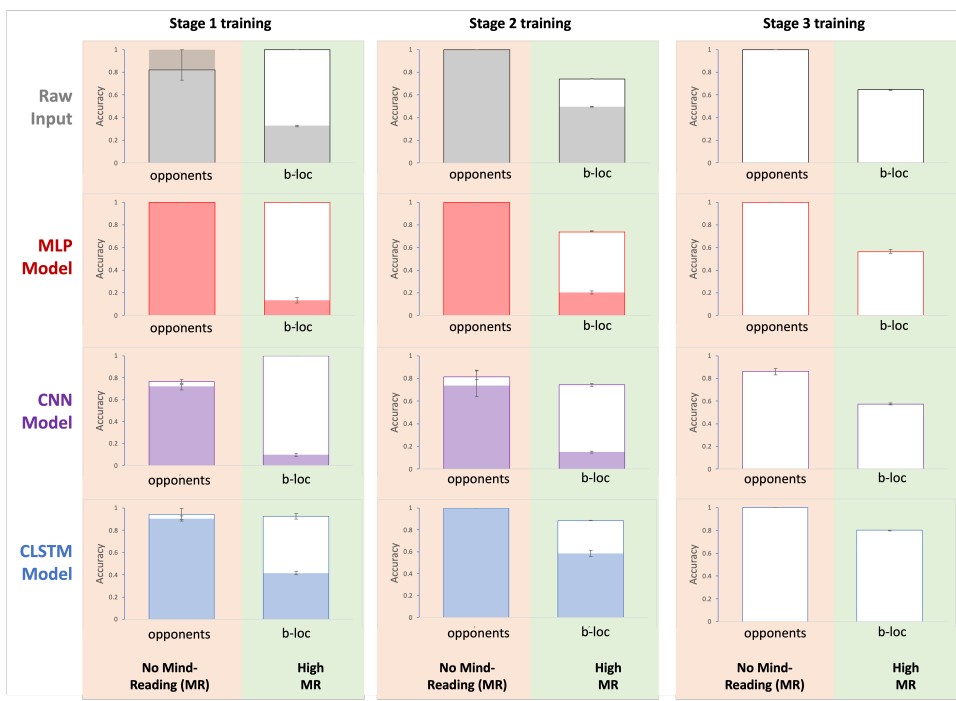

Figure 1: Results of Experiment 1 for all models using MLP feature extraction. Mean IFR scores on familiar tasks (tasks in the given dataset) are depicted using outlined bars, while filled bars represent mean IFR scores on novel tasks. Error bars indicate the quartiles of each value across the nine extraction-models.

its ability to handle sequential data seems to give it a clear advantage over the simpler CNN and the MLP.

In this experiment, we additionally recorded results for the final layers of the networks alone. We find that, within these activations, representations are weaker than the sum total of all layers.

## 4 EXPERIMENT 2: DIFFERENTIATING HYPOTHESES ABOUT TOM REPRESENTATIONS

In this experiment, we examine three hypotheses about the nature of our models' ToM capabilities. In order to evaluate these three hypotheses empirically, we first enumerate a list of features whose presence we may quantify in our models. Next, we calculate the degree to which each of these features is represented in our models, as in Experiment 1, and discuss what that evidence implies regarding the models' learned strategies for solving the task.

### 4.1 REPRESENTATION STRATEGIES

Similar to Horschler et al. Horschler et al. (2023), we consider a small number of mindreading strategies, ordered by their representation power. We refer to these three strategies as non-mindreading, low-mindreading, and high-mindreading. These are inspired by psychologists' theories about the evolution and development of ToM in humans and primates, in particular Butterfill & Apperly (2013), though we refrain from using the same terminology in order to make specific claims about what computational features each strategy allows for.

No-mindreading strategies are the hypothesis that a model does not make inferences about the mental states of others. When using these strategies, a model would only ever solve a task by learning surface-level statistics about the environment and the perceived behavior of others. These strategies are closely associated with behavior reading, outlined by Heyes (Heyes, 2015). A no-mindreading

model is incapable of accounting for an opponent's knowledge and beliefs, except for where those states happen to correlate with correct solutions in its past experience. Such correlations could be purely coincidental, so they are not meaningful in novel circumstances.

Low-mindreading strategies represent a more advanced level of social cognition, in which limited inferences might be made about other agents. In this framework, the subject is capable of understanding that an agent could have perceptions and goals that are different to its own. Similarly to both minimal ToM and perception-goal psychology, these states must be grounded in reality: the other's perceptions are a part of the visible world-state, and the other's goal is a concrete world state that the other will attempt to reach. While this rudimentary mentalization is not capable of making inferences about counterfactual beliefs, it does allow for an understanding that the opponent is or is not aware of some factual information.

High-mindreading refers to the ability to attribute abstract mental states to others. This representation allows for the inference of specific counterfactual beliefs, and is capable of replacing concrete goals with abstract desires, e.g. instead of reasoning that an opponent is going to go to *that treat over there*, we might reason that the opponent wants the largest available treat, wherever it may be. High-mindreading models might be likened to agents with full-blown ToM.

## 4.2 SELECTED FEATURES

| Name | Description | Example | Interpretation |
| --- | --- | --- | --- |
| **prediction** | the task-model's selection | [1, 0, 0, 0, 0] | box 0 is predicted |
| **opponents** | the number of opponents | [1] | there is an opponent present |
| **big-loc** | the large treat locations | [1, 0, 0, 0, 0] | large treat is at box 0 |
| **small-loc** | the small treat location | [1, 0, 0, 0, 0] | small treat is at box 0 |
| **vision** | whether the opponent can see | [1, 0, 0, 0, 0] | opponent's vision is obscured in timestep 0 |
| **fb-exist** | opponent false belief about existence | [1, 0] | opponent is misinformed about small treat existence |
| **fb-loc** | change-of-location false belief | [1, 0, 1, 0, 0] | opponent is misinformed about treats at box 0 and 2 |
| **b-loc** | opponent beliefs about treat location | [[1, 0], [0, 1], …] | opponent believes small treat is at box 0, etc. |
| **target-loc** | opponent's current target | [1, 0, 0, 0, 0] | opponent's goal is location 0 |
| **labels** | the correct box for the player | [1, 0, 0, 0, 0] | box 0 is correct |

Table 1: Descriptions of each feature that we select to extract from the activations of our task-models. Aside from vision, which uses all five timesteps, features are sampled from the last timestep of the environment.

To distinguish the three strategies, we identify a set of ten features, and hypothesize that each will only be accurately represented in either familiar or unfamiliar tasks depending on the strategy taken. These features are described in Table 1.

For non-mindreading models, we anticipate that *predictions*, the task-model's outputs, should be represented well. This hypothesis is a sanity check for the IFR extraction technique, as it makes the assumption that features which would be available to or predictable by the task-models should likewise learnable by the extractor-models. The only reason that predictions would not generalize well to novel tasks is if surprising inputs resulted in unexpected task-model behavior. Stage-1 includes every event sequence in the environment, including all variations aside from opponent presence, so novel tasks (which would only be novel by the introduction of an opponent) do not contain other surprising observable features. We also anticipate the potential for strong representation of the treat locations, which describes the location of either treat. The treats' positions are never uncertain from the perspective of the player. Even a non-mindreading player who has learned the larger treat will often be claimed by an opponent might track both treats' positions in case the opponent is absent. Per the results of Experiment 1, opponent presence is easy to track and generalizes well to novel scenarios in all models.

For low-mindreading models, we anticipate the introduction of *vision*, a five-length vector which describes whether the opponent's vision was obscured during any given timestep. We also expect *target-loc*, the opponent's goal, to be well-represented for familiar cases, but not necessarily for novel cases. These models should also represent *fb-exist*, a vector which describes whether the opponent harbors a false belief about either of the two treats' existence. This is because low-mindreading models are capable of tracking a lack of awareness, but they cannot necessarily recall whether an opponent harbors a specific counterfactual belief.

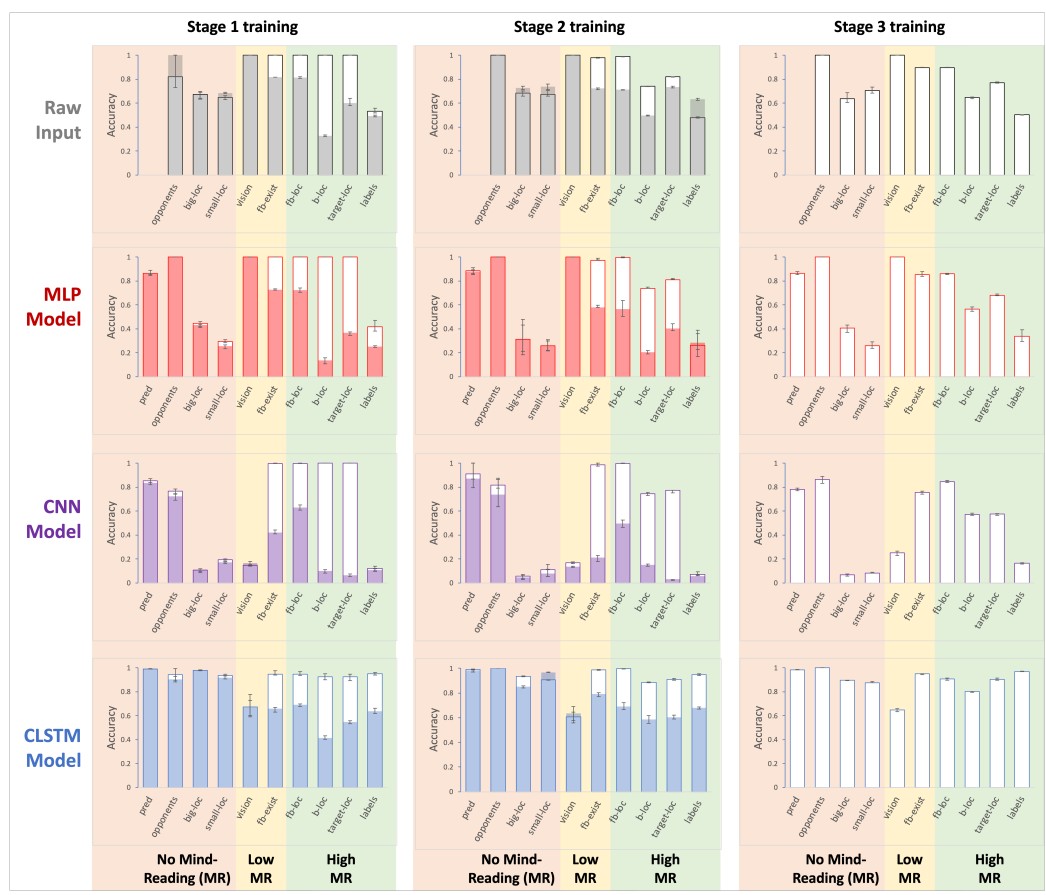

Figure 2: Results of Experiment 2 for all models using MLP feature extraction. Mean IFR scores on familiar tasks (tasks in the given dataset) are shown as outlined bars, while filled bars represent mean IFR scores on novel tasks. Error bars indicate the quartiles of each value across the nine extraction-models. Features are grouped by strategy: no mindreading, low mindreading, and high mindreading. Because *predictions* is model-dependent, it cannot be inferred from raw input.

Finally, for high-mindreading models, we predict robust representation of *b-loc*, which describes the opponent's beliefs about the locations of both treats (quantified in Experiment 1). We additionally predict strong representation of *fb-loc*, a five-length vector that describes counterfactual beliefs. *Fb-loc*'s values are all zero, except for locations where the opponent believes a treat is in the wrong location and is correct regarding that treat's existence. Because high-mindreading models can reason about false beliefs, *goal-loc* should robustly generalize in their results. Finally, given goal-loc and the locations of both treats, high-mindreading models should robustly predict *labels*, the correct outputs that the task-models are trained to predict.

## 4.3 RESULTS

The four non-mindreading features are represented surprisingly poorly by the CNN and MLP models. *predictions*, which functions as a sanity check for the IFR scoring technique, is well-captured by all our tested models. Both MLP and CNN models score poorly for the features that describe the treats' true locations, across all datasets. Of the three strategies, non-mindreading features have by far the most robust generalization to novel tasks, across all models and datasets. Only the CLSTM model represents the non-mindreading features well, surpassing the input IFR scores on all datasets.

While the low-mindreading features are not represented well by the CNN model, they are represented well by the MLP and CLSTM models. For all models and datasets, *vision*, an observable feature of the environment, robustly transfers to novel tasks, but *fb-exist*, a feature that must be

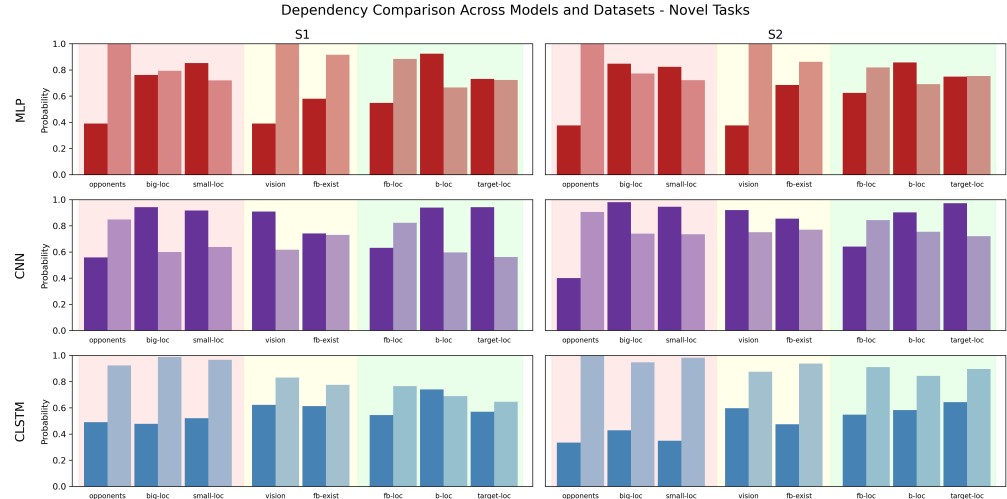

Figure 3: Experiment 3 results on novel tasks. Values of $P(A|F)$, that the task-model will be accurate given that a feature is accurately represented, are shown in darker bars. To the right of each of those, in lighter color, are $P(\neg A|\neg F)$ values. As with the previous two experiments, features are grouped by strategy.

synthesized from other information, does not. The high-mindreading features have more varied performance. *Fb-loc* is well-represented on familiar tasks for all models but those trained on S3. All models trained on S1 represent *fb-loc*, *b-loc*, and *target-loc* well on familiar tasks, but generalize those features poorly. Only the CLSTM learns to represent *labels* on familiar tasks, but it does not generalize that feature well to novel tasks.

## 5    EXPERIMENT 3: FEATURE REPRESENTATIONS AND TASK ACCURACY

We have shown how features may be measured in model activations, but we have yet to examine the relationship between feature representation and task accuracy. In addition to our predictions about which features should be present in models of different representation strategies, we also predict that those features make meaningful contributions to the model's outputs. Such contributions might exist even for novel tasks. We may examine them by investigating how model accuracy depends on any given feature: if, on a given datapoint, the feature is accurately represented, does that imply that the task-model's prediction will be accurate? Inversely, if the feature is inaccurately represented, will the task-model be inaccurate? We may represent these two quantities as probabilities, $P(A|F)$ and $P(\neg A|\neg F)$, the probability that the model will be Accurate given that a Feature is present, and the corresponding case when the Feature is absent.

In this experiment, we are investigating when those two accuracy values coincide, as opposed to a causal relationships between them. As such, this technique does not directly measure sufficiency and necessity; findings should be regarded as evidence for sufficiency and necessity instead.

### 5.1    RESULTS

Among all the features tested, none have overall negative effects on accuracy, supporting their relevance to the task. Interestingly, while many features exhibit evidence of either necessity or sufficiency, few provide both, highlighting the complex nature of features in this task. In the CLSTM model, most tested features appear to be more necessary than they are sufficient, particularly in the models trained on S2. This suggests that either combinations of these features, or additional features not in our list, are required for robust generalization to novel tasks. Several such features are necessary for almost every datapoint in the evaluation set. In the CNN model, there exist several features that are more sufficient than they are necessary. This implies that there are multiple viable strategies for robust generalization that the CNN model is able to employ. The single feature which

provides the most sufficiency across all models is *(b-loc)*. This result is reasonable, as *b-loc* encodes the opponent's goal, and often might contain information about the other treat's location as well. Its prominence suggests that understanding the opponent's beliefs about treat locations is a key factor in successful task completion, regardless of model architecture.

# 6 DISCUSSION AND LIMITATIONS

## 6.1 DISCUSSION

In general, we find that the best-generalizing features present in our models are those that do not require mindreading. Models that represent low- and high-mindreading features do perform better on the task overall, however, and all features in our set aid prediction accuracy when represented well. As for the strategies used by our selected models architectures, the CNN and MLP models do not even succeed at no-mindreading strategies on familiar tasks. The CLSTM models are adept at no-mindreading strategies, but they are unable to robustly generalize low- and high-mindreading features to novel tasks. While the training regime has a strong effect on the features represented for familiar tasks, its effect on novel task features is weak relative to that of model architecture, suggesting that model engineering could be more important for learning robust ToM skills than learning curricula.

## 6.2 FEATURE EXTRACTION

Our feature extraction technique is subject to causes of both over- and under-performing, producing scores that differ from models' true ability to represent various features. Over-performance could be caused by the introduction of nonlinearity to the models. Because we assume features might be difficult to uncover within a connectionist task-model's inner state, we require some amount of nonlinearity in our extractor-models. But that same nonlinearity could be leveraged to synthesize information that the task-model ignores. Additionally, the task itself likely contains spurious correlations that could be abused by the extractor-model. Under-performance could be caused by both over- and under-fitting (over-fitting only in the case of novel-task performance). We used identical training for all task-models, despite their inputs and outputs having different shapes and distribution, so certain features are likely underfit. Additionally, the task-models often feature layers that perform significantly more complex operations than the extractor-models, so they might synthesize the information more easily. Overall, feature extraction using extractor-models is a measure of only one piece of evidence that any feature is actually represented by the task-model.

## 6.3 ACCURACY DEPENDENCY

Our technique for determining accuracy dependency itself depends on feature extraction, so it is subject to the same limitations. Its direct comparison of accuracy on individual datapoints allows for a more fine-grained view of features' contributions, but these effects could likewise include spurious correlations.

## 6.4 FUTURE WORK

With minor environmental extensions, further mechanisms could be tested using the same methodology that we have presented, by identifying and quantifying differentiating features, e.g. those included in theorized mechanisms that reason about uncertainty, preferences, and multiple agents. The work presented in this paper is intended to serve as a precursor to the engineering and investigation of models which embody mechanistic hypotheses about ToM skills. Only by understanding the specific features that those models represent do we believe we can empirically validate models' robustness to varied tasks. In addition to handcrafted mechanisms for mentalization, we intend to evaluate contemporary machine learning techniques including transformer architectures. The grid-world used in this paper is implemented primarily as a reinforcement learning environment, so we also plan to examine the effect of embodied cognition, where the player is an agent with their own goals and actions in which environmental knowledge may be grounded.

## REPRODUCIBILITY STATEMENT

The code and instructions to produce the results in this paper can be found in a github link, omitted for review.

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
