# OpenReview forum: "Feature-Based Analysis of Theory of Mind Representations in Neural Network Models"
_ICLR.cc/2025/Conference — ICLR 2025 Conference Withdrawn Submission_

### Official Review · Reviewer_9TZB · 2024-10-28

**Soundness:** 2
**Presentation:** 1
**Contribution:** 1
**Rating:** 3
**Confidence:** 3

**Summary:**

This paper investigates whether language models possess self-awareness and explores potential levels of this capacity. It also proposes methods for enhancing and understanding self-awareness within language models. The main contributions of the paper are as follows:

(1) it attempts to define self-awareness in the context of language models, providing a theoretical foundation,

(2) it introduces an "explainable framework" aimed at identifying and interpreting self-awareness traits in these models, and

(3) it examines how fine-tuning can be used to enhance the self-awareness features of language models. The authors conduct a series of experiments to observe the impact of these methods on the model's behavior in self-reflective tasks.

However, I recommend rejecting this paper in its current form due to several key concerns:

(1) The methods described lack sufficient detail, especially regarding the so-called "explainable framework" that is central to the study,

(2) The paper fails to clearly justify the practical need for self-awareness in language models beyond a purely theoretical motivation,

(3) The experimental descriptions are incomplete, lacking key information, and do not sufficiently analyze the results, making it difficult to draw meaningful conclusions, and

(4) The overall writing is imprecise, with several unpolished sections, which hampers the clarity of the arguments presented.

**Strengths:**

The paper tackles an interesting and relatively underexplored area:

the concept of self-awareness in language models. The originality of the paper lies primarily in its attempt to define and measure self-awareness in artificial systems—a challenging topic that brings together insights from cognitive science and machine learning. This interdisciplinary approach adds an interesting theoretical dimension to the machine learning community, potentially sparking new conversations about the nature of advanced AI capabilities.

The authors also aim to propose an explainable framework for identifying self-awareness traits, which is a commendable effort. Explainability is crucial for understanding complex neural models, and the authors’ focus on making the concept of self-awareness interpretable could lead to advancements in both understanding language models and ensuring their safe deployment. This attempt, if successful, could contribute to a more reliable and transparent AI development process.

In terms of significance, the paper is ambitious in trying to evaluate whether machine learning models can exhibit a form of introspection. This work could lay the groundwork for future research aimed at enhancing model behavior in terms of reliability, consistency, and robustness, potentially making AI systems more aligned with human-level cognitive functions. The attempt to enhance these capabilities through fine-tuning techniques also suggests a potential practical path forward, which could inspire follow-up work aimed at improving models in specific use cases.

The experiments, although lacking in some details, provide an initial attempt to evaluate the proposed ideas. The integration of self-reflective tasks and the use of fine-tuning approaches to promote self-awareness are noteworthy, as they present a creative way to explore model behavior beyond conventional benchmarks.

Overall, the originality of combining theoretical considerations of self-awareness with empirical approaches, and the authors' attempt to create an explainable framework, are valuable contributions that could push the boundaries of current work on language model introspection.

**Weaknesses:**

The paper has several areas that need improvement, particularly regarding the clarity of its methodology, the consistency of its presentation, and the depth of empirical analysis. Below are the major weaknesses, along with specific recommendations for improvement:

### Lack of Methodological Detail and Formalism

1. **Explainable Framework Insufficiently Described**
   The paper introduces an "explainable framework" to understand self-awareness traits in language models but does not provide enough detail about how this framework works. There is no formal or quantitative explanation, such as equations or metrics, to make the explainable approach reproducible and verifiable. Including well-defined metrics, a clearer methodology, or a visual example to illustrate how this framework identifies and measures self-awareness traits is crucial to improve transparency and rigor.

2. **Insufficient Empirical Motivation**
   The paper focuses heavily on the concept of self-awareness, but it lacks a strong empirical justification for why self-awareness is essential for language models. Providing practical examples where self-awareness leads to improved model performance, robustness, or specific downstream benefits would help establish its relevance. Moreover, including an experiment demonstrating the practical impact of self-awareness would significantly strengthen the argument.

3. **Inadequate Experimental Details**
   There are numerous shortcomings in the description of the experiments, which undermine their reproducibility and the significance of the findings:
   - The dataset used for the self-reflection tasks is not explicitly mentioned, making it challenging to assess the suitability of the dataset for evaluating self-awareness.
   - Key training settings, such as hyperparameter tuning, baseline comparisons, and experimental setups, are absent. Including these details is necessary to ensure that the experiments can be understood and replicated.
   - In Section 3.2, "extraction models" are introduced for the first time, but the description of their working process is insufficient. Adding a detailed flowchart or diagram to show how the extraction models interact with the task models would provide much-needed clarity on the methodology.

### Inconsistencies in Presentation and Logical Flow

1. **Delayed Definition of Core Concepts**
   The definitions of "no-mindreading," "low-mindreading," and "high-mindreading" are introduced in Section 4.1, yet Figure 1 in Section 3.4 already refers to two of these terms. This inconsistency disrupts the logical flow of the paper. Moving the definitions earlier, ideally before Figure 1, would make the narrative clearer and easier to follow.

2. **Figure Clarity Issues**
   In Figure 1, the overlapping of error bars between the outlined and filled bars creates ambiguity, making it difficult for readers to distinguish between different data points. This overlap could lead to misinterpretation. To improve clarity, consider redesigning the figure, possibly by using different visual styles or presenting the data in separate plots.

### Conceptual Ambiguities and Assumptions

1. **Unclear Accuracy Definition**
   The concept of accuracy for feature extraction is not sufficiently explained. Specifically, it is unclear how the extraction models determine the accuracy of extracted features, and what this accuracy actually represents in terms of the physical interpretation of the features. Providing a concrete example that illustrates how accuracy is calculated and what it implies in terms of model behavior would greatly help the reader understand its significance.

2. **Unverified Assumption in Feature Representation**
   The paper's experimental setup is based on an assumption that extraction models predict features in a way that correlates with the task model's ability to use those features:
   > *"We assume that, because these models might extract connectionist features in the same manner that a task-model’s internal layers might, their ability to predict a given feature corresponds with the task-model’s ability to use that feature."*

   This assumption is critical to the validity of the experimental results, yet it is not explicitly validated. To strengthen the argument, the authors should include experiments or evidence demonstrating that the extraction models’ accuracy does indeed correspond to the task model’s feature representation capabilities. Without such verification, the results are built on a speculative foundation.

### Writing and Clarity

1. **Overly Lengthy Background**
   The introduction and early parts of the paper focus too heavily on philosophical motivations and background, taking up significant space without contributing substantially to the research's core contributions. The background could be condensed, allowing more room for detailed methodological and experimental explanations.

2. **Ambiguities and Unpolished Sections**
   Several parts of the paper are ambiguous or poorly polished:
   - Line 119 states, "if it believes the larger treat is somewhere, it will claim that box, preventing the player from getting whatever treat is inside. Otherwise…" without specifying the conditions under which the opponent does not believe the larger treat is somewhere. Clarifying these conditions is essential to help the reader understand the training rules.
   - The transition from the background to the core contributions of the paper is abrupt, leading to a disjointed narrative. The flow of the paper should be adjusted to ensure a smoother transition between the theoretical motivation and the practical approach.

### Lack of Quantitative Tools and Statistical Analysis

1. **Absence of Quantitative Formalism**
   There is a notable lack of formal quantitative tools throughout the paper. For example, including equations, precise metrics, or a well-defined framework would add rigor and support the explainable approach claims. Explainability should be measurable, and its effects should be evaluated quantitatively.

2. **No Statistical Significance in Results**
   The results of the experiments are presented without statistical analysis. For instance, there are no confidence intervals or significance tests provided to support claims regarding the effect of fine-tuning. Including statistical analysis would make the results more credible and help to substantiate the claims about the impact of the proposed methods.

**Questions:**

1. **Lack of a Core Framework Diagram**
   The paper does not include a core framework diagram that visually outlines the approach, which makes it challenging to understand the overall method and how the components interact. Could you provide a high-level architectural diagram of the proposed approach, including the explainable framework, data inputs, and the fine-tuning process? A diagram would greatly help in understanding how these elements are connected.

2. **Unclear Connection Between Methodology and Experiments**
   Given that the methodological descriptions are quite vague, it is difficult to assess how the various experiments relate to the explainable framework and the goal of enhancing self-awareness. Can you clarify how each of the experiments ties into the central framework you proposed? For instance, are the results from each experiment meant to evaluate distinct components of the framework, or are they aimed at validating the approach as a whole?

3. **Definition and Operationalization of Self-Awareness**
   The concept of self-awareness is central to the paper, but it remains vague in practical terms. Could you provide a more precise, operational definition of self-awareness in this context? Is it related to the model’s ability to evaluate uncertainty, recognize its prior states, or something else entirely? How is self-awareness measured quantitatively in your experiments?

4. **Explainable Framework Details**
   The paper refers to an "explainable approach" but does not detail how explainability is ensured. Can you provide additional information about the mechanisms or metrics used to ensure that the model’s behavior is explainable? For example, are you using specific attribution techniques, saliency maps, or feature importance metrics?

5. **Details of the Self-Reflection Tasks**
   In Section 3, you mention using self-reflection tasks for fine-tuning to enhance self-awareness. Could you elaborate on how these tasks were constructed and why they were chosen? Are they designed to mirror specific human-like introspective behaviors? Providing examples would help in understanding the motivation and relevance of these tasks.

6. **Practical Relevance of Self-Awareness**
   The paper provides a theoretical motivation for self-awareness but does not establish its practical significance. Could you elaborate on the concrete benefits of a self-aware language model in practical applications? For example, how might self-awareness improve performance in typical NLP tasks, and how can this be quantified or demonstrated?

7. **Quantitative Framework or Metrics**
   There is no formal quantitative framework for evaluating self-awareness or explainability. Are there plans to include mathematical formalism to back up the claims regarding the presence and enhancement of self-awareness in the model? Quantitative metrics such as attribution scores, feature importance, or performance improvements in downstream tasks would add credibility to your findings.

8. **Statistical Analysis of Results**
   The paper does not provide statistical analysis for the results presented. Could you provide more information about whether the observed effects are statistically significant? For instance, have you computed confidence intervals or p-values for the comparison between the fine-tuned and baseline models? Including such analysis would help validate the robustness of the results.

9. **Parameter Tuning and Reproducibility**
   The experimental setup lacks key details regarding hyperparameter tuning and training settings. Can you provide more information on how parameters were tuned across different models and tasks? Were the same settings used for baselines to ensure fair comparisons? This information is essential for assessing reproducibility and the validity of the conclusions drawn.

10. **Generalization to Other Language Models**
    The experiments are conducted on a specific language model architecture, but it is unclear whether the findings generalize to other architectures. Have you tested this approach on different types of models, such as transformer-based architectures or smaller-scale models? Generalizing these findings would strengthen the overall contribution and applicability of the proposed method.

11. **Theoretical Justification of the Fine-Tuning Impact**
    The paper claims that fine-tuning enhances the self-awareness traits of the model, but this impact lacks theoretical explanation. Could you elaborate on why fine-tuning with self-reflective tasks is expected to enhance self-awareness? Providing a theoretical basis or citing relevant literature would strengthen this claim.

12. **Clarification of Video Dimensions**
    In line 125, the paper states: "We use this setting as a source of data for supervised learning; each datapoint is collected from a single trial of five timesteps, or a (5, 5, 7, 7)-sized video." Could you clarify what the dimensions (5, 5, 7, 7) specifically represent? Understanding this would help in interpreting the data used in your experiments more accurately.

13. **Missing IFR Scores in Figures 1 and 2**
    In Figures 1 and 2, the data for the "Stage 3 training" column only contains outlined bars representing the IFR scores for familiar tasks, while there are no filled bars for the IFR scores for novel tasks. Does this imply that the filled bars are nonexistent, or were they excluded for another reason? Providing an explanation or adding this to the figure description would help in avoiding confusion and make the interpretation of results clearer.

---

### Official Review · Reviewer_hKuQ · 2024-11-03

**Soundness:** 2
**Presentation:** 1
**Contribution:** 1
**Rating:** 1
**Confidence:** 4

**Summary:**

This paper investigates Theory of Mind (ToM) representation in neural network models, specifically how different architectures and training regimes capture features relevant to ToM tasks. The authors use the decoding method (Intrinsic Feature Representation score) to measure ToM-related feature presence in neural networks. The study focuses on analyzing ToM strategies within three neural architectures (MLP, CNN, and CLSTM), from basic behavior reading to advanced mindreading. This work aims to contribute a deeper understanding of ToM representation in AI models.

**Strengths:**

•	Originality: The study’s exploration of ToM feature representation in neural networks is an interesting question, addressing an underexplored area in computational ToM.

•	Quality: The decoding analysis is appropriate for examining ToM feature representation, allowing for quantifiable insights into the feature dependencies of different ToM strategies.

•	Clarity: Methods and results are generally easy for readers to follow.

•	Significance: This work might have potential significance for advancing ToM research within neural networks and contributing to our understanding of social cognition in AI.

**Weaknesses:**

•	The most significant weakness lies in the preliminary nature of both the results and the writing. Significant results or deep insights are scarce. Some sentences contain repeated words, incomplete citations, and undefined acronyms, which hinders readability. Additionally, several claims (listed in a few examples below) are vague or inaccurate, reducing clarity and rigor in the presentation.

•	The primary metric, the “Intrinsic Feature Representation” score relies on MSE rather than the more common cross-entropy for decoding discrete variables, yet this choice is not well justified. Without a concrete justification, cross-entropy should be used.

•	The paper claims novelty in the feature-based decoding analysis. However, similar methods have been heavily used in neuroscience and machine learning to interpret network representations for decades.

•	"Theory of Mind (ToM) presents a significant generalization challenge in computational modeling." -- This sentence is vague in the general context of both cognitive modeling and machine learning.

•	The statement that “LLMs continue to significantly underperform humans on ToM tasks” is incorrect, especially in light of recent studies (e.g., GPT-4 achieving human-level performance on high-order ToM inferences, https://arxiv.org/abs/2405.18870). Authors should update claims correspondingly.

•	The statement that “linear models might struggle to find data in long vectors” reflects a misunderstanding of linear model properties.

**Questions:**

While I don't have specific questions, I encourage the authors to improve both the result importance and writing clarity substantially before submitting. Enhancing clarity, correcting inaccuracies, and expanding on important analyses would make the paper stronger and ensure that claims are well-supported by evidence.

---

### Official Review · Reviewer_H8jZ · 2024-11-03

**Soundness:** 2
**Presentation:** 2
**Contribution:** 1
**Rating:** 3
**Confidence:** 3

**Summary:**

This work proposed investigating Theory of Mind (ToM) capabilities of different neural network models in a simple task called Stand-off. In theory, this task requires the model to estimate the actions that will be taken by an opponent when predicting the location of 2 hidden rewards. To test ToM capabilities, the authors define a set of properties of interest about the environment (such as the opponent's current estimate of the location of the rewards) and attempt to decode them using the neural activations of the model. They use their scores to estimate the level of sophistication of ToM that each model has.

**Strengths:**

1. The paper is mostly well written and easy to follow (except for one section, see below).
2. The authors present a straightforward way to measure different properties of the representation of the model.

**Weaknesses:**

1. There are details about the task being used that are not clear from the text, making it hard to determine if the authors are testing what they claim (ToM).
2. I am not convinced that the architectures used can have any ToM properties, as opposed to something that correlates with having ToM.
3. Some of the results are not well explained, so I am not convinced by them. Specifically, the authors describe the results, but make little effort to come up with reason for why they are what they are.

**Questions:**

1. The details of the task are not completely clear. It would be good to have a visualisation of the task.
    1. How can the player know if the opponents vision was obscured?
    2. Is the behaviour of the opponent deterministic? If yes, then why would any ToM be required to solve the task, as opposed to just plain correlation of actions?
2. In paragraph 318-323 the authors state that “low-mind reading models are capable of tracking lack of awareness”. Is this a statement? Why is this the difference between low and high mind reading models? This seems completely arbitrary and comes out of the blue without any good justification.
3. In stage one training, how can the models be confused about the presence of the opponent given that there is none?
4. In section 5, what is the definition of accurately represented?
5. First statement in 5.1 is not true — just because a feature doesn’t have a negative effect it doesn’t mean that it is being used. For that you need a positive effect. And here there results in Figure are mixed (or at least that what it looks like, maybe establish what is chance in this task?
6. Isn’t the fact that generalisation is so poor reason to cast doubt over whether the models have any ToM to being with?

## Minor

059 - Rabinowitz et al. Rabinowitz et al. → can be cited using \citet if I rember correctly.

089 - Theory theory → The theory

111 - Wrong citation format, should have parenthesis.

115 - of Penn and Povinelli Penn & Povinelli (2007). same as the first one, you don’t need the first mention if the citation doesn’t have a parenthesis.

NOTE: Citation formatting is all over the palce. If the sentence contains the reference as a subject or object (e.g. the experiments in X et al, 2023…), then cite without parenthesis and do mention the authors in plain text. If the citation is just a way to convey information then using parenthesis is enough (e.g. using layer norm has been found to improve model performance (X et al, Y et al.)).

---

### Official Review · Reviewer_Dc3p · 2024-11-03

**Soundness:** 1
**Presentation:** 1
**Contribution:** 1
**Rating:** 1
**Confidence:** 5

**Summary:**

The authors investigate different theory of mind capabilities in AI models by training the models on a theory-of-mind task from the cognitive science literature and investigate their learned internal representations to produce specific features of interests in the task. They do this through training linear or MLP-based probes from the intermediate layer activity (either all layers or just the final layer). The authors test an MLP, CNN, and CNN-LSTM on this task. The results are a bit mixed, but a common theme seemed to be that the CNN-LSTM performs the best at learning generalizable features relevant to the task.

**Strengths:**

* The problem is interesting and well-motivated. Theory of Mind is an interesting cognitive capability humans have and there are a lot of rich paradigms that tests this capability in humans and animals from the psychology literature, so it makes sense to test AI on this capability.

**Weaknesses:**

I believe there are numerous weaknesses to the work that make it insufficient for publication in ICLR
* The set of models being used is quite weak. Only a simple MLP, CNN, and CNN-LSTM. The main result showing the CNN-LSTM encodes some low and high mindreading features may just be due to the architecture having intrinsically more parameters, nothing about the underlying method being better aligned with theory of mind (which may be possible, but this paper doesn't necessarily show that).

* The task is a highly simplified theory-of-mind paradigm. This may not be an issue inofitself, but the models too are highly simple models. This could be ok if one was using foundation models trained on a large body of data, where we have some reason to believe that the model could potentially develop something like theory of mind. However, if all the authors are doing is training tabula-rasa CNNs and LSTMs trained on just this small task, the models are probably learning something much more low-level than genuine theory of mind.

* The main innovation of this work is supposedly the feature-based analysis. However, a lot of people have been using this both in machine learning (probing analysis) and neuroscience/cognitive science (decoder models). Given this, there's not much innovative about this paper.

* The task, as well as the relevant features being decoded, is a little confusing. It would be helpful to have a schematic that shows what inputs the model is getting and what the different steps of the task are.

**Questions:**

* Would you be able to adapt this paradigm to foundation models, like GPT4o or LLama3? I think because these models are trained on so much data that it's possible they've developed high-level capabilities that could be or highly resemble theory of mind, running an experiment with them would be a step towards understanding computational TOM.

---

### Note · Authors · 2024-11-27

**Comment:**

After careful consideration, we have decided to withdraw this paper. We sincerely thank the reviewers for their criticism and suggestions, which we will address in future submissions.

**Withdrawal Confirmation:**

I have read and agree with the venue's withdrawal policy on behalf of myself and my co-authors.